# Factors Associated with Smoking Behaviors in Out-of-School Youth: Based on an Ecological Model

**DOI:** 10.3390/ijerph18126380

**Published:** 2021-06-12

**Authors:** Hye-Young Song, Sook-Ja Yang

**Affiliations:** 1Department of Nursing, Woosuk University, Jeonbuk 55338, Korea; lemonbam84@woosuk.ac.kr; 2College of Nursing, Ewha Womans University, Seoul 03760, Korea

**Keywords:** out-of-school youths, smoking, ecological model, adolescents

## Abstract

Purpose: To investigate the association between smoking behavior in out-of-school youths (OSY) and individual, interpersonal, and organizational factors through an ecological model. Methods: Participants were 297 OSY aged 13–18 years, who visited J area’s counseling center. The independent variables were self-control (intrapersonal factor), parental attachment and social network (interpersonal factors), and exposure to no-smoking policy (community factor). The dependent variable was smoking. Descriptive statistics, χ^2^-tests, correlation analyses, and logistic regression analysis were performed. Results: The predictors of smoking in OSY were analyzed using demographic, intrapersonal, interpersonal, and community factors. Period after discontinuation of school, self-control, parental attachment, and friends’ smoking behavior were significant predictors. Smoking behavior was found to decrease when the period after discontinuation of school was over than one year, when the parents attachment increased and when self-control increased. Conversely, smoking increased when friends’ smoking increased. Conclusions: When administering smoking prevention programs for OSY, parental attachment and psychological traits, such as self-control, should be taken into consideration. Therefore, for effective results, such programs need to strengthen self-control, stress the importance of parental attachment through parent education, and enroll adolescents along with their friends who smoke. Implications and Contribution: Based on an ecological model, this descriptive survey, conducted to investigate the association between smoking behavior among out-of-school youths and individual, interpersonal, and organizational factors, proposes that smoking prevention programs should consider parental attachment and psychological traits, including self-control, for optimal effectiveness.

## 1. Introduction

The number of Korean students in elementary, middle, and high schools who drop out has been rising every year: 47,663 in 2016 (0.8% of the total student population) to 52,539 in 2018 (0.9% of the total student population) [1]. With regard to the youths’ overall school dropout rates, Korea is still in a better situation compared with America–7% of its adolescents’ dropout rates [2]. However, Korea has shown its increasing rates and with inclusion of working youths, it is estimated to have a larger size in this field [3,4].

As OSY are not within school boundaries, their families are the ones who can likely control their smoking [1]. However, this rarely happens as OSY tend to live away from their families or have little family involvement [3]. Counselors in youth counseling organizations have difficulty prioritizing smoking cessation as adolescents face serious concerns and challenges, including family problems and violence [5].

Previous studies have reported that the smoking rate is higher among OSY than in-school adolescents [1,2,3,4,5]. The percentage of adolescents smoking an average of 10 cigarettes a day was found to be 57.9% and 19.6% among OSY and in-school adolescents, respectively [3]. Further, the percentage of adolescents who have purchased cigarettes at a convenience store was found to be 79.8% and 47.0% among OSY and in-school adolescents, respectively. Compared to in-school adolescents (61.9%), only 23.4% OSY receive smoking cessation education, indicating that education about smoking cessation is lacking in the context of OSY [2].

Smoking initiation at an early age intensifies nicotine dependence, which increases the likelihood of becoming a lifelong smoker and substantially shortens lifespan [6]. It is also associated with a higher prevalence of chronic obstructive pulmonary disease, coronary artery disease, respiratory diseases, and other detrimental effects, such as weakening of lung function [7]. Therefore, preventing smoking initiation among adolescents is crucial.

Smoking behaviors in adolescents are associated with genetic, social, and environmental factors. Individuals’ health behaviors are determined by their relationships with family or colleagues, physical environment, and personal characteristics [8]. Therefore, individuals’ social and physical environments must be examined [9]. An ecological model is beneficial in enabling multidimensional and diverse interventions, as the predictors of health behaviors can be analyzed through each component in the model [10].

Most studies on smoking trends have been conducted with adolescents in school; they invariably examined intrapersonal and interpersonal factors [11,12]. Few studies have examined the predictors of smoking in OSY from a multidimensional perspective. This study aims to address this gap by investigating factors associated with smoking behaviors in OSY based on an ecological model, with a focus on intrapersonal, interpersonal, and community factors. The findings can be used as foundational data for developing intervention strategies to promote smoking cessation in OSY and for devising related policies.

## 2. Objectives

We aimed to identify factors associated with smoking behaviors in OSY based on the ecological model proposed by McLeroy et al. [10]. Intrapersonal (self-control), interpersonal (parental attachment, social network), and community (no-smoking policy exposure) factors were established. The specific objectives were as follows:

1. Examine the demographic characteristics and intrapersonal, interpersonal, and community factors in OSY.

2. Compare the differences in smoking behaviors according to demographic characteristics and intrapersonal, interpersonal, and community factors in OSY.

3. Identify the demographic characteristics and intrapersonal, interpersonal, and community factors associated with smoking behaviors in OSY. 

## 3. Methods

### 3.1. Study Design

A descriptive survey was used in the study.

### 3.2. Study Participants

The study population was the OSY of an adolescent counseling and welfare center in J Province in 2020. In adolescent welfare and counselling centers, out-of-school youth are offered a preparation guide for the College Scholastic Ability Test, career development programs and healthy behavior program. The sample size was determined using G*Power 3.1 software. We conducted a logistic regression analysis with a medium effect size (odds ratio [OR] = 1.5, H0 = 0.2, X parm µ = 3, X parm σ = 1), significance of 0.05, power of 0.90, and a two-tailed test [13]. The minimum sample size was 277. In consideration of potential withdrawals, 300 questionnaires were collected. Three questionnaires (1% withdrawal rate) with incomplete responses were excluded. The final analysis included 297 questionnaires. Participants included individuals aged 13–18 years who had not received formal middle or high school education in at least six months and had visited the adolescent counseling and welfare center in J Province at least once.

### 3.3. Study Instruments

We identified existing instruments appropriate for the construct through a literature review and focus-group interviews. The instruments were modified for use in this study. The content validity of 17 items on demographic characteristics, 20 items on the Self-control Scale (intrapersonal factor), 9 items on the Important People Instrument (version for smoking), 25 items on the Inventory of Parent attachment (interpersonal factor), and 7 items on no-smoking policy exposure (community factor) were evaluated by a panel comprising four nursing professors and two OSY experts over two rounds of assessment. A pilot study was conducted with 30 OSY from 15 January to 30 January 2020, to ensure the survey was comprehensible and to determine how much time would be required to complete it. The questionnaire consisted of 78 items and took approximately 20 minutes to complete. The questionnaire was easy to understand, without any awkward phrasing.

### 3.4. Intrapersonal Factor

#### Self-Control

Adolescents’ self-control was measured using the Self-control Scale developed by Gottfredson and Hirschi [14] and modified and adapted by Nam and Ok [15]. This 20-item scale consists of 10 items for long-term satisfaction seeking and 10 items for instant satisfaction seeking. Long-term satisfaction measures the ability to concentrate and to delay one’s desires in order to effectively solve problems. Instant satisfaction measures impulsivity, egocentric thinking, and the tendency to act before thinking or talking.

### 3.5. Interpersonal Factors

#### 3.5.1. Parental Attachment

Parental attachment was measured using the modified Korean version of the Inventory of Parent and Peer Attachment developed by Yoo and colleagues [16], by translating the parent attachment subscale of the original inventory developed by Armsden and Greenberg [17]. This 25-item tool consists of 10 items for trust, 9 for communication, and 6 for alienation. Each item is rated on a 5-point Likert scale, ranging from 1 (*never true*) to 5 (*always true*). Ten items (3, 6, 8, 9, 10, 11, 14, 17, 18, and 23) were reverse scored. Total scores range from 25 to 125, with higher scores indicating stronger parental attachment. Cronbach’s αs were 0.94 for paternal attachment in a previous study [16] and 0.95 in this study.

#### 3.5.2. Social Network

Social networks related to adolescent smoking were measured using the Important People Instrument (version for smoking) [18] after modifying the tool. It consists of 10 items, but one item about the effectiveness of smoking cessation therapy was deleted because it was irrelevant to our objective. Item 1 instructed participants to list up to 10 important people in their social networks. Item 2 instructed participants to indicate their relationships with these important people. Items 3–9 asked questions about these important people. Items 3, 6, and 7 asked about the frequency of contact with the important people, smoking intensity and smoking frequency of these people. These items were measured as continuous variables. Items 4, 5, 8, and 9 measured bonding with these people, frequency of social support, smoking support, and no-smoking support, respectively. Items 4, 5, 8, and 9 were rated on a 5-point Likert scale from 1 (*never*) to 5 (*always*). This tool consists of social network traits, social support, and involvement of smokers in the social network. The formula for computing the factors of these subscales is shown below:

##### Social Network Traits

Social network traits consist of the size of the social network, the proportion of friends, and average contact frequency. Social network size refers to the number of important people in the social network, scored 1–10. The proportion of friends is the percentage of friends in the social network based on the total number of people in the network. The average contact frequency was calculated by measuring the contact frequency with each member of the social network (times/day) and dividing this by the total number of people in the social network.

##### Social Support

Social support consists of average general social support, social support frequency, smoking support, no-smoking support, and quality of ties. The average general support was calculated by dividing the frequency of support by the total number of people in the social network. The range of average social support was between 1 and 5. The frequency of social support was calculated by multiplying the contact frequency with each member (times/day) and social support and adding the values for all members. Social support was rated from 1 (*strongly disagree*) to 5 (*strongly agree*). A higher frequency of support indicates greater social support from an important person. 

Support related to smoking is the sum of smoking support and no-smoking support provided by each member of the social network (5-point scale). Smoking support and no-smoking support were rated from 1 (*strongly disagree*) to 5 (*strongly agree*). A higher score for smoking support indicates greater smoking support by people in the social network. Similarly, a higher score for no-smoking support indicates greater no-smoking support. The quality of ties is the sum of the contact frequency (times/day) with each member of the social network multiplied by the importance of the member. A higher score indicates a better quality of ties. 

##### The Involvement of Smokers in Social Networks

The involvement of smokers in the social network refers to smokers in the social network, friends and family that smoke, and the degree of smoke infiltration by members in the social network. Smokers in the social network are the percentage of smokers, percentage of friends who smoke, and percentage of family members that smoke in the total number of people in the social network. The degree of smoke infiltration by members of the social network is the sum of the contact frequency (times/day) with each member of the social network multiplied by their smoking intensity (number of cigarettes per day).

### 3.6. Community Factor

#### Exposure to Smoking Cessation Policy

The public policies of the smoking section in the 14th Korea Youth Risk Behavior Web-based survey [19] were used after adaptation. The yes-or-no questions about public policies on smoking were revised using a 5-point Likert scale. Exposure to smoking cessation policy was rated from 1 (*strongly disagree*) to 5 (*strongly agree*). The no-smoking policy exposure was measured using seven items. The total score ranged from 7 to 35, with a higher score indicating greater no-smoking policy exposure. Cronbach’s α was 0.81 in this study.

### 3.7. Data Collection

The study was approved by E University Institutional Review Boards (approval no. 202001-0013-02). The participants were informed of the study’s purpose, duration, procedure, voluntary participation, matters pertinent to protection of personal information and data disposal, and the freedom to withdraw consent without any disadvantages. Completed questionnaires were inserted in a sealed envelope and were collected in a box placed inside the institution. The questionnaire data does not contain personally identifiable information such as name and contact information, and they were assigned personal ID numbers and were stored as encrypted electronic data. The questionnaires and electronic data will be retained for 3 years following the completion of the study, after which they will be permanently destroyed and discarded.

Parent consent was obtained for underage adolescents as per the guidelines of the institutional review board. If adolescents wished to participate, but could not obtain their parents’ consent, they gained it from their counselors at the welfare centers. Data were collected from 15 February to 31 March 2020, from adolescents at the adolescent counseling and welfare center. The study recruitment advertisement was posted on a bulletin board at the welfare center. For the pilot study, the questionnaires, caregiver and participant consent forms were sent to the homes of the consenting adolescents in a sealed envelope by the counselors at the adolescent counseling and welfare center. Counselors at the center sent out a packet consisting of the questionnaire, caregiver and participant consent form to those who consented to participate. The completed forms were sealed in the envelope and dropped into a designated box at the center, which were then collected for analysis.

### 3.8. Data Analysis

Collected data were analyzed with IBM SPSS statistics 25.0 (IBM, New-York, NY, USA). Significance was set at 5%. The differences in smoking behaviors according to demographic characteristics, self-control, parental attachment, social network, and no-smoking policy exposure were analyzed using χ2- and *t*-tests. The correlations among participants’ self-control, parental attachment, social network, and no-smoking policy exposure were analyzed using Pearson’s correlation coefficients. The effects of participants’ demographics, parental attachment, social network, and no-smoking policy exposure on smoking behaviors were analyzed using a logistic regression analysis and presented as ORs and 95% confidence intervals (CIs). A four-stage logistic regression model was used. Model 1 contained demographic characteristics; while intrapersonal, interpersonal, and community factors were additionally entered in Models 2, 3, and 4, respectively.

## 4. Results

### 4.1. Differences in Smoking Behaviors Based on Demographic Characteristics

The differences in smoking behaviors based on participants’ demographic characteristics are presented in Table 1. The mean age was 16.01 ± 1.50 years in the smoker group, which was lower than that in the non-smoker group (*p* < 0.001). A greater percentage of OSY dropped out of middle school in the smoker group than in the non-smoker group (*p* = 0.003). The percentage of OSY with less than 1 years of schooling was higher in the smoker group than in the non-smoker group (*p* < 0.001). The percentage of OSY who dropped out of school owing to family matters was higher in the smoker group than in the non-smoker group (*p* = 0.001). The percentage of OSY who dropped out of school to hang out with friends outside the school was higher in the smoker group than in the non-smoker group (*p* < 0.001). The percentage of OSY who dropped out of school owing to conflict with parents was higher in the smoker group than in the non-smoker group (*p* < 0.001). The percentage of OSY who used to have poor grades was higher in the smoker group than in the non-smoker group (*p* < 0.001).

### 4.2. Smoking Behaviors Based on Intrapersonal, Interpersonal, and Community Factors

Smoking behaviors according to intrapersonal, interpersonal, and community factors are shown in Table 2. The mean self-control score was higher in the non-smoker group than in the smoker group (*p* < 0.001). Concerning interpersonal factors, the parental attachment score was higher in the non-smoker group than in the smoker group (*p* < 0.001). Concerning social network, the smoking group had a higher proportion of friends (*p* < 0.001), average contact frequency (*p = 0*.002), social support frequency (*p = 0*.013), smoking support (*p* < 0.001), quality of ties (*p* = 0.002), smokers in the social network (*p* < 0.001), friends’ smoking (*p* < 0.001), and involvement of smokers in the social network (*p* < 0.001) compared with the non-smoker group. Conversely, the non-smoker group had higher general social support (*p* = 0.009) and no smoking support (*p* < 0.001). Concerning community factors, no-smoking policy exposure was higher in the smoker group than in the non-smoker group (*p* = 0.007).

Self-control, parental attachment, proportion of friends, average contact frequency, average general social support, social support frequency, smoking support, no-smoking support, quality of ties, social network smokers, friend smokers, the social network smoker’s involvement, and no-smoking policy exposure were statistically significant in smoking behaviors.

### 4.3. Correlations among Intrapersonal, Interpersonal, and Community Factors

The correlations among independent variables are presented in Table 3. Strong correlations were seen between (a) average contact frequency and social support frequency and quality of ties; (b) social support frequency and quality of ties; and (c) smokers in one’s social network and friends’ smoking, based on which multicollinearity is expected. Thus, average contact frequency, social support frequency, smokers in the social network, and quality of ties were excluded from the logistic regression model.

### 4.4. Factors Associated with Smoking Behaviors

To identify the predictors of smoking behaviors, a four-stage logistic regression analysis was performed by including significant variables. Although proportion of friends and non-smoking support was significant, we excluded them as they are constructs that include friends and are relatively less associated with smoking. No-smoking support and smoking support were mutually complementary constructs; therefore, we excluded no-smoking support owing to our objective. The results are presented in Table 4.

Model 1 contained significant demographic characteristics: age, period after discontinuation of school, and school achievement. The odds for smoking behaviors decreased by 46% with increasing age and by 85% with one year or longer off school compared to less than one year off school. The odds for smoking increased by approximately 275% with 1 unit decreased in school achievement.

The intrapersonal factor (self-control) was added to Model 2. The odds for smoking decreased by 23% with 1 unit increase in self-control. Interpersonal factors (parental attachment, average general social support, smoking support, friend smokers, social network smoker’s involvement) were added to Model 3. The odds for smoking decreased by 8% with 1 unit increase in parental attachment and increased by 4% with 1 unit increase in friends’ smoking.

The final model (Model 4) containing demographic characteristics, intrapersonal factors, interpersonal factors, and community factors (no-smoking policy exposure) indicated that the odds for smoking were 85% lower when away from school for one year or longer as compared to less than one year. The odds for smoking decreased by 15% with 1 unit increase in self-control, by 7% with 1 unit increase in parental attachment, and it increased by 4% with 1 unit increase in friends’ smoking.

## 5. Discussion

Using an ecological approach as the conceptual basis, we established four models to identify factors associated with smoking behaviors in OSY. We examined key relationships by establishing Model 1 with demographic characteristics and adding intrapersonal, interpersonal, and community factors in Models 2, 3, and 4, respectively. 

The final model (Model 4) showed that period after discontinuation of school, self-control, parental attachment, and friends’ smoking were associated with smoking behaviors. OSY with one year or longer off school were less likely to smoke than their counterparts who had been off school for less than one year. Soon after dropping out, adolescents experience a spike in their stress levels as they adjust to the new environment and impulsively express their emotions through smoking; however, over time, they may build coping strategies against stress and adapt to their situation. Once addicted to smoking, education or promotion on quitting smoking is much less effective; hence, it is extremely important to provide education on smoking cessation to adolescents before addiction occurs [20]. Therefore, smoking rate should be lowered in these adolescents by implementing smoking prevention programs during fresh dropout (within one year) to increase their awareness about the perils of smoking [21]. Smoking rate should be lowered in these adolescents by implementing smoking prevention and smoking cessation programs during this period to increase their awareness about the perils of smoking.

Smoking rate increases with decreasing self-control, a finding similar to that from previous studies [22,23]. People with low self-control tend to have low concentration and are less able to postpone their desire to deal with problems efficiently before acting [24]. Such a lack of self-control tends to cause maladaptation or delinquency, such as smoking and drinking in adolescence [22]. Lane and colleagues [25] noted self-control problems as a cause of smoking among adolescents.

Parental attachment, an interpersonal factor, was also found to be associated with smoking [26,27]. Parental attachment predicted problem behaviors in adolescents [28]. A study on the common factors between smoking and delinquency also found that parenting behavior (i.e., emotional bonding between parents and children) was the most potent predictor of smoking in adolescents. Consistent with our findings, parental attachment has a grave impact on social behavioral development and children’s psychological and emotional functions [29].

The odds of smoking increased with an increase in the number of friends who smoked within the social network. Influences one’s family members exert on them decrease as they age; thus, their peers become their bigger influencers at puberty [30]. Peer groups serve as important role models for sociality and behavioral development during adolescence. They function as a socialization group, similar to family and school [31]. Friends’ smoking significantly predicted one’s initiation to and continuation of smoking [32].

The community factor (no-smoking policy exposure) was added to Model 4, but its association with smoking in OSY was non-significant. A previous study found that acceptance of a no-smoking policy increases with increasing exposure to the policy [33,34]. The success of an enforced policy depends on the level of acceptance of the target population [32]. Subsequent studies should focus on developing an instrument for Measuring degree of exposure to no-smoking policies that reflect variations in the acceptance of these policies among OSY.

This study has several implications. First, we identified multidimensional factors associated with smoking in OSY using an ecological model. Second, we presented theoretical evidence to focus on interpersonal factors when developing smoking prevention programs for OSY. These findings will be useful as foundational data for developing smoking prevention programs for OSY.

Nonetheless, this study has limited generalizability because the sample was restricted to a single region. Subsequent studies should include participants from diverse regions to analyze the predictors of smoking. Additionally, based on an ecological model, we confirmed that period after discontinuation of school, self-control, parental attachment, and friends’ smoking are associated with smoking in OSY. Subsequent studies should develop smoking prevention interventions for OSY and assess the effectiveness of these programs. Furthermore, future studies should utilize objective indices (e.g., carbon monoxide concentration, nicotine test, cotinine test), as opposed to self-report questionnaires, for a more accurate measure of the degree of smoking.

## 6. Conclusions 

Fresh dropouts (within one year off school), self-control, parental attachment, and friends’ smoking should be considered when developing smoking prevention and smoking cessation programs for OSY.

When implementing smoking prevention programs for OSY, it is important to note that OSY showed poorer parental attachment and personal psychological characteristics such as self-control compared to their in-school counterparts. Thus, programs that boost self-control, parent education programs stressing the importance of parental attachment, and smoking cessation education programs that enroll smokers along with their friends are needed for OSY.

## Figures and Tables

**Table 1 ijerph-18-06380-t001:** Smoking behaviors according to demographic characteristics.

Variable		Total (n = 297)	Smoker (n = 187)		Non-Smoker (n = 110)		χ^2^ or t	*p*	
		n	n	%	n	%			
		Mean ± SD	Mean ± SD		Mean ± SD				
Age		16.39 ± 1.41	16.01 ± 1.50		17.05 ± 0.90		−6.59	<0.001	
Gender	Men	220	134	74.1	86	71.7	78.2	1.54	0.215
	Women	77	53	25.9	24	28.3	21.8		
Discontinuation of school	Middle school	160	113	53.9	47	60.4	42.7	8.73	0.003
	High school	137	74	46.1	63	39.6	57.3		
Period after discontinuation		1.72 ± 1.26	1.54 ± 1.39		2.04 ± 0.93		−3.69	<0.001	
	<1 year	62	54	20.9	8	28.9	7.3	19.57	<0.001
	≥1 year	235	133	79.1	102	71.1	92.7		
Reasons for discontinuation (multiple responses)	Difficulty in studying	147	87	49.5	60	46.5	54.5	1.78	0.182
	Lack of study needs	130	89	43.8	41	47.6	37.3	3.00	0.083
	Violation of school regulations	66	45	22.2	21	24.1	19.1	0.99	0.319
	Economic status of family	38	33	12.8	5	17.6	4.5	10.66	0.001
	Out-of-school friends’ interaction	67	59	22.6	8	31.6	7.3	23.37	<0.001
	Conflict with parents	59	50	19.9	9	26.7	8.2	14.98	<0.001
	Other	1	1	0.3	0	0.5	0.0	0.59	0.442
School achievement	High, Medium	131	60	44.1	71	32.1	64.5	29.60	<0.001
	Low	166	127	55.9	39	67.9	35.5		
Residing with (multiple responses)	Parents	290	181	97.6	109	96.8	99.1	1.59	0.207
	Grandparents	57	34	19.2	23	18.2	20.9	0.33	0.564
	Sister, Brother	232	148	78.1	84	79.1	76.4	0.31	0.576
Allowance (10,000 won)		5.47 ± 3.09	5.37 ± 2.84		5.64 ± 3.48		−0.67	0.504	
	<5	180	117	60.6	63	62.6	57.3	0.81	0.367
	≥5	117	70	39.4	47	37.4	42.7		
Age at which smoking was started(age)			14.27 ± 1.61						
	10–12	29	29	15.5	15.5				
	13–15	112	112	59.9	59.9				
	16–18	46	46	24.6	24.6				
Number of years of smoking			1.67 ± 1.20						
	<1	53	53	28.3	28.3				
	1–2	42	42	22.5	22.5				
	2–3	54	54	28.9	28.9				
	≥3	38	38	20.3	20.3				
Daily smoking quantity (number of cigarettes)			7.51 ± 5.71						
	<5	50	50	26.7	26.7				
	5–9	81	81	43.3	43.3				
	≥10	56	56	29.9	29.9				
Number of smoking days in 1 month			19.82 ± 11.71						
	1–2	25	25	13.4	13.4				
	3–5	19	19	10.2	10.2				
	6–9	13	13	7.0	7.0				
	10–19	19	19	10.2	10.2				
	20–29	20	20	10.7	10.7				
	daily	91	91	48.7	48.7				

**Table 2 ijerph-18-06380-t002:** Smoking behaviors based on intrapersonal, interpersonal, and community factors.

Variable	Total (n = 297)		Smoker (n = 187)	Non-Smoker (n = 110)	t	*p*
	Range	M ± SD	M ± SD	M ± SD		
Intrapersonal Factors						
Self-control	20–100	57.96 ± 10.94	52.23 ± 7.58	67.69 ± 8.68	−16.08	<0.001
Interpersonal Factors						
Parental attachment	29–125	73.77 ± 13.51	67.29 ± 9.12	84.80 ± 12.62	−12.73	<0.001
Social network						
Size	1–10	2.09 ± 1.62	1.94 ± 0.95	2.35 ± 2.34	−1.74	0.087
Friends (%)	0–100	82.24 ± 33.91	89.39 ± 26.39	70.08 ± 41.20	4.42	<0.001
Average contact frequency (day)	0–10	1.77 ± 1.53	1.99 ± 1.46	1.41 ± 1.57	3.30	0.002
Social support						
Average general social support	1–5	3.93 ± 0.80	3.85 ± 0.83	4.09 ± 0.72	−2.64	0.009
Social support frequency	5–50	7.08 ± 6.35	7.78 ± 6.12	5.89 ± 6.58	2.50	0.013
Smoking support	1–50	5.29 ± 3.97	6.56 ± 3.60	3.14 ± 3.62	7.90	<0.001
No smoking support	1–50	7.05 ± 8.05	4.78 ± 3.58	10.91 ± 11.40	−5.49	<0.001
Quality of ties	5–35	7.21 ± 6.17	8.04 ± 5.97	5.78 ± 6.27	3.09	0.002
Smoking involvement						
Social network smokers (%)	0–100	62.34 ± 45.15	89.57 ± 24.43	16.04 ± 32.88	20.38	<0.001
Friend smokers (%)	0–100	54.63 ± 47.20	82.89 ± 32.03	6.58 ± 24.52	23.06	<0.001
Family smokers (%)	0–100	7.49 ± 22.99	6.68 ± 22.96	8.87 ± 23.08	−0.79	0.429
Social network smokers’ involvement	0–200	7.47 ± 15.31	10.27 ± 11.65	2.72 ± 19.20	4.22	<0.001
Community Factors						
Exposure to non-smoking policy	7–35	23.28 ± 3.91	23.75 ± 3.53	22.49 ± 4.40	2.70	0.007

**Table 3 ijerph-18-06380-t003:** Correlation among intrapersonal, interpersonal, and community factors.

Variable	Self-Control	Parental Attachment	Friend Rate (%)	Average Contact Frequency (days)	Average Social General Support	Social Support Frequency	Smoking Support	No-Smoking Support	Quality of Ties	Social Network Smokers (%)	Friend Smokers	Social Network Smokers’ Involvement	Non-Smoking Policy Exposure	
**Intrapersonal**	Self-control	1												
Interpersonal	Parental attachment	0.604 **	1											
	Friend rate (%)	−0.214 **	−0.294 **	1										
	Average contact frequency (day)	−0.199 **	−0.068	0.139 *	1									
	Average social general support	0.072	0.208 **	0.072	0.075	1								
	Social support frequency	−0.197 **	−0.025	0.141 *	0.959 **	0.298 **	1							
	Smoking support	−0.363 **	−0.231 **	0.131 *	0.061	0.158 **	0.105	1						
	No-smoking support	0.316**	0.353**	−0.251**	−0.113	0.139*	−0.079	0.261*	1					
	Quality of ties	−0.248 **	−0.066	0.142 *	0.968 **	0.167 **	0.962 **	0.108	−0.123 *	1				
	Social network smokers (%)	−0.572 **	−0.497 **	0.193 **	0.194 **	−0.049	0.184 **	0.391 **	−0.371 **	0.196 **	1			
	Friend smokers	−0.583 **	−0.533 **	0.521 **	0.218 **	−0.011	0.218 **	0.449 *	−0.380 **	0.235 **	0.871 **	1		
	Social network smokers’ involvement	−0.119 *	−0.086	0.046	0.577 **	0.079	0.583 **	0.449 *	−0.133 *	0.503 **	0.384 **	0.305 **	1	
Community	Exposure to non-smoking policy	−0.062	−0.113	0.033	−0.135 *	0.079	−0.130 *	0.222 **	0.150 **	−0.114 *	0.169 **	0.182 **	−0.146 *	1

* *p* < 0.05, ** *p* < 0.01.

**Table 4 ijerph-18-06380-t004:** Effects of variables on smoking behavior.

Variable		Model 1				Model 2				Model 3				Model 4			
		OR	95% CI		*p*	OR	95% CI		*p*	OR	95% CI		*p*	OR	95% CI		*p*
			LLCI	ULCI			LLCI	ULCI			LLCI	ULCI			LLCI	ULCI	
General	age	0.545	0.430	0.692	<0.001	0.566	0.383	0.837	0.004	0.867	0.480	1.567	0.636	0.850	0.459	1.573	0.604
	Discontinuation of school (period) (1 year more, ref = 1 year or less)	0.147	0.063	0.343	<0.001	0.088	0.029	0.273	<0.001	0.143	0.030	0.672	0.014	0.147	0.030	0.709	0.017
	School achievement (Low, ref = High, Medium)	3.757	2.148	6.572	<0.001	1.236	0.557	2.738	0.603	1.585	0.491	5.119	.441	1.597	0.494	5.166	0.434
Intrapersonal	Self-control					0.774	0.726	0.825	<0.001	0.852	0.782	0.927	<0.001	0.851	0.782	0.927	<0.001
Interpersonal	Parental attachment									0.924	0.861	0.992	0.030	0.927	0.861	0.998	0.043
	Average social general support									0.662	0.291	1.508	0.326	0.655	0.286	1.499	0.317
	Smoking support									1.029	0.879	1.203	0.725	1.026	0.877	1.201	0.748
	Friend smokers									1.037	1.023	1.051	<0.001	1.037	1.022	1.051	<0.001
	Social network smoker’s involvement									1.030	0.997	1.064	0.075	1.031	0.997	1.066	0.079
Community	Non-smoking policy exposure													1.019	0.870	1.194	0.817
		χ^2^ = 85.83, df = 3, (*p* < 0.001) Nagelkerke R^2^ = 0.344				χ^2^ = 222.89, df = 4, (*p* < 0.001) Nagelkerke R^2^ = 0.723				χ^2^ = 297.12, df = 9, (*p* < 0.001) Nagelkerke R^2^ = 0.863				χ^2^ = 297.17, df = 10, (*p* < 0.001) Nagelkerke R^2^ = 0.863			

## Data Availability

The data presented in this study are available on request from the corresponding author. The data are not publicly available due to the protection of the privacy of research subjects.

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
