# Peer review of "Factors Associated with Smoking Behaviors in Out-of-School Youth: Based on an Ecological Model"

_ijerph, 2021, doi:10.3390/ijerph18126380_

Round 1
Reviewer 1 Report
This paper presents findings from a survey of 297 adolescents ages 13-18 who were currently not attending school in a region of South Korea. The survey assessed various characteristics of the youth, their family relations, and their social networks with a focus on their cigarette smoking. The findings generally replicated what is found for in-school youth who smoke, namely that they have weaker self-control than non-smokers, poorer relations with parents, and more association with peers who smoke. One unique finding was the authors claim that smokers were more likely to have spent less time out of school than their non-smoking peers. But this does not seem to match what is in Table 1, where it shows that non-smokers were more likely to have been out of school for less than 1 year. It does not easily follow therefore that one should intervene early with out-of-school youth in order to prevent smoking. It could just be that out-of-school youth who smoke drop out of school earlier than those who do not smoke. In addition, it appears that OSY were smoking even before they left school. The advice that it makes sense to intervene early is probably correct, but there is nothing in these data to support that conclusion.
Other conclusions could also be modified. Youth with self control problems may have difficulty making healthy decisions, but one would not characterize this as “frequently making mistakes.” It is an over-statement to say that “friends, and not family members have the greatest influence on adolescents…” Parents still have a lot of influence but friends grow in importance as adolescents age. And there is nothing in the data to show that parents have less influence than peers.
It is stated in the Results that “The odds for smoking increased by approximately 375% for poor grades…” when it is more correct to say that they increased by 275%. It would also help to make clear that the regressions in Table 4 are for the odds of being a smoker.
For readers who are unfamiliar with the J region of South Korea, it would help to define what this means. It would also be helpful to know more about why the youth recruited for the study were visiting the counselling center, and whether those who smoked were visiting the smoking cessation part of the center.
Some wording could be improved. The scales used to measure frequency were not Likert scales. Those are agree-disagree scales. Also, there is no formula presented on page 3 for computing the social network scores.
Author Response
Point 1: One unique finding was the authors claim that smokers were more likely to have spent less time out of school than their non-smoking peers. But this does not seem to match what is in Table 1, where it shows that non-smokers were more likely to have been out of school for less than 1 year. It does not easily follow therefore that one should intervene early with out-of-school youth in order to prevent smoking. It could just be that out-of-school youth who smoke drop out of school earlier than those who do not smoke.
Response 1: page 5-6.
Because the configuration in Table 1 made it difficult for you to read the values, we corrected it to display the number of samples (N) first, followed by percentage (previous versions showed percentage first, and then sample counts).
Meanwhile, we believe you may have had some difficulties in understanding our calculating results because we computed them based on the vertical sum of 100%, not a horizontal sum of it.
- The smoking rate of youth who have been out of school for less than 1 year is 54/62=87.1%
- The smoking rate of youth who have been out of school for more than 1 year is 133/235=56.6%
OSY has a higher smoking rate (87.1%) than those who have been out of school for more than a year (56.6%).
In line with this circumstance, the welfare and counselling centers provide them a few smoking prevention programs. Yet, they have been equipped with few studies regarding comparison of smoking rates before and after the teens’ using smoking cessation part of the centers. Given all this, we cannot be 100% sure, but can judge carefully that the adolescents’ smoking rates may have dropped. For more solutions, we will follow-up this issue with our future studies.
Point 2: In addition, it appears that OSY were smoking even before they left school. The advice that it makes sense to intervene early is probably correct, but there is nothing in these data to support that conclusion
Response 2: page 9.
We revised according to your feedback.
As a basis for our opinions, we have included some contents of a report below.
“Once addicted to smoking, education or promotion on quitting smoking is much less effective; hence, it is extremely important to provide education on smoking cessation to adolescents before addiction occurs [20, 23].”
Reference) USDHHS(U.S. Department of Health and Human Services). Preventing Tobacco Use Among Young People: A Report of the Surgeon General. (1994). Atlanta, Georgia: U.S. Department of Health and Human Services, Public Health Service, Centers for Disease Control and Prevention, National Center for Chronic Disease Prevention and Health Promotion, Office on Smoking and Health.
Point 3: Other conclusions could also be modified. Youth with self control problems may have difficulty making healthy decisions, but one would not characterize this as “frequently making mistakes.” It is an over-statement to say that “friends, and not family members have the greatest influence on adolescents…” Parents still have a lot of influence but friends grow in importance as adolescents age. And there is nothing in the data to show that parents have less influence than peers
Response 3: page 9-10.
We revised according to your feedback.
In addition, we tried not to go too far in expressing some situations.
- People with low self-control frequently make mistakes, are defiant, and have a quick temper
-> There is a tendency to believe that people with low self-control lack concentration and that they do not think carefully before acting.
- The influence of family members declines with advancing age; thus, friends and not the family members, have the greatest influence on adolescents
-> Influences one’s family members exert on them decrease as they age; thus, their peers become their bigger influencers at puberty.
Point 4: It is stated in the Results that “The odds for smoking increased by approximately 375% for poor grades…” when it is more correct to say that they increased by 275%. It would also help to make clear that the regressions in Table 4 are for the odds of being a smoker.
Response4: page 9
We revised according to your feedback.
The odds for smoking increased by approximately 275% with 1 unit decreased in school achievement
Point 5: For readers who are unfamiliar with the J region of South Korea, it would help to define what this means
Response 5:
The youths’ school dropout rates in Jeollabukdo province are 0.5%, which is lower than 0.9%, Korea’s average ones. Yet, students in the region are dropping out of school at an increasing rate. This is one of the reasons why we selected Jeollabukdo province for this topic
Reference) [4] Lee, J, Y,; Choi, J, H. Jeollabukdo out of school youth survey and support plan. Jeonbuk institute, 2019.
Point 6: It would also be helpful to know more about why the youth recruited for the study were visiting the counselling center, and whether those who smoked were visiting the smoking cessation part of the center
Response 6: page 2
In welfare and counselling centers for adolescents, out-of-school youths are offered a preparation guide for the College Scholastic Ability Test and career development and healthy behavior programs. In line with this circumstance, the welfare and counselling centers provide them a few smoking prevention programs. Yet, they have been equipped with few studies regarding comparison of smoking rates before and after the teens’ using smoking cessation part of the centers. Given all this, we cannot be 100% sure, but can judge carefully that the adolescents’ smoking rates may have dropped.
Point 7: Some wording could be improved. The scales used to measure frequency were not Likert scales. Those are agree-disagree scales.
Response 7: page 4
The exposure to smoking cessation policy had Yes or No questions in it first, but we converted them into a 5 point scales.
Ex) In the last 30days, it was difficult to buy cigarettes at convenience stores.
1=strongly disagree
2=little disagree
3=moderate
4=little agree
5= strongly agree
Point 8: Also, there is no formula presented on page 3 for computing the social network scores
Response 8: If you look at the bottoms of page3 and page4, you will find we have presented the formula of each social network variables.
1) Social network traits
Social network traits consist of the size of the social network, the proportion of friends, and average contact frequency. Social network size refers to the number of important people in the social network, scored 1–10. The proportion of friends is the percentage of friends in the social network based on the total number of people in the network. The average contact frequency was calculated by measuring the contact frequency with each member of the social network (times/day) and dividing this by the total number of people in the social network.
(2) Social support
Social support consists of average general social support, social support frequency, smoking support, no-smoking support, and quality of ties. The average general support was calculated by dividing the frequency of support by the total number of people in the social network. The range of average social support was between 1 and 5. The frequency of social support was calculated by multiplying the contact frequency with each member (times/day) and social support and adding the values for all members. Social support was rated from 1 (strongly disagree) to 5 (strongly agree). A higher frequency of support indicates greater social support from an important person.
Support related to smoking is the sum of smoking support and no-smoking support provided by each member of the social network (5-point scale). Smoking support and no-smoking support were rated from 1 (strongly disagree) to 5 (strongly agree). A higher score for smoking support indicates greater smoking support by people in the social network. Similarly, a higher score for no-smoking support indicates greater no-smoking support. The quality of ties is the sum of the contact frequency (times/day) with each member of the social network multiplied by the importance of the member. A higher score indicates a better quality of ties.
(3) The involvement of smokers in social networks
The involvement of smokers in the social network refers to smokers in the social network, friends and family that smoke, and the degree of smoke infiltration by members in the social network. Smokers in the social network are the percentage of smokers, percentage of friends who smoke, and percentage of family members that smoke in the total number of people in the social network. The degree of smoke infiltration by members of the social network is the sum of the contact frequency (times/day) with each member of the social network multiplied by their smoking intensity (number of cigarettes per day).

Reviewer 2 Report
The Authors investigated the association between smoking behavior in out-of-school youths (OSY) and individual, interpersonal, and organizational factors through an ecological model.
- While this is a critical topic, the stats about the demographic, presented in the introduction, suggests that this may only be of interest to other demographics, which makes the limitations highlighted by the authors even more significant.
- On the abstract, the statement on the "duration of discontinuation of studies" is hard to follow.
- Not sure that objective 2 and 3 present completely different information.
- Please provide more information on the sample size calculation as the use of the software mentioned, with the given information, provides a minimum sample size of 409. It might be that the sample size is correct, but the test parameters are not all mentioned.
- Editorial input needed, for example, in page 5, the fourth line is missing a 'period'. In page 7, there is 'friendstwork t'. There is also 'logestic regression'. Check the entire document for errors.
- Explain the results before each Table. This should be elaborated better for the readers to get a clear idea of what is the most significant combination of factors on the different models.
- Since OSY is influenced by many factors, it would have been great if authors included the most common factor, which is economic background. Are the youths dropping out of school due to financial constraint more prone to smoking behavior.
Author Response
Point 1: While this is a critical topic, the stats about the demographic, presented in the introduction, suggests that this may only be of interest to other demographics, which makes the limitations highlighted by the authors even more significant.
Response 1: page1
We revised according to your feedback.
The number of Korean students in elementary, middle, and high schools who drop out has been rising every year: 47,663 in 2016 (0.8% of the total student population) to 52,539 in 2018 (0.9% of the total student population) [1]. With regard to the youths’ overall school dropout rates, Korea is still in a better situation compared with America – 7% of its adolescents’ dropout rates[2]. However, Korea has shown its increasing rates and with inclusion of working youths, it is estimated to have a larger size in this field[3-4].
Reference) Lee, J, Y,; Choi, J, H. Jeollabukdo out of school youth survey and support plan. Jeonbuk institute, 2019.
Point 2: On the abstract, the statement on the "duration of discontinuation of studies" is hard to follow.
Response 2: page1
We revised according to your feedback.
Smoking behavior was found to decrease when the period after discontinuation of school was over than one year, when the parents attachment increased and when self-control increased.
Point 3: Not sure that objective 2 and 3 present completely different information.
Point 4: Please provide more information on the sample size calculation as the use of the software mentioned, with the given information, provides a minimum sample size of 409. It might be that the sample size is correct, but the test parameters are not all mentioned.
Response 4: page2
We revised according to your feedback.
We conducted a logistic regression analysis with a medium effect size (odds ratio [OR]=1.5, H0=0.2, X parm µ=3, X parm σ=1), significance of .05, power of 0.90, and a two-tailed test [13].
Point 5: Editorial input needed, for example, in page 5, the fourth line is missing a 'period'. In page 7, there is 'friendstwork t'. There is also 'logestic regression'. Check the entire document for errors.
Response 5: We revised according to your feedback.
We have corrected the editorial errors.
Point 6: Explain the results before each Table. This should be elaborated better for the readers to get a clear idea of what is the most significant combination of factors on the different models.
Response 6: page5-6
We revised according to your feedback.
Differences in smoking behaviors based on demographic characteristics
The differences in smoking behaviors based on participants’ demographiccharacteristics are presented in Table 1. The mean age was 16.01 ± 1.50 years in the smoker group, which was lower than that in the non-smoker group (p < .001). A greater percentage of OSY dropped out of middle school in the smoker group than in the non-smoker group (p = .003). The percentage of OSY with less than 1 years of schooling was higher in the smoker group than in the non-smoker group (p < .001). The percentage of OSY who dropped out of school owing to family matters was higher in the smoker group than in the non-smoker group (p = .001). The percentage of OSY who dropped out of school to hang out with friends outside the school was higher in the smoker group than in the non-smoker group (p < .001). The percentage of OSY who dropped out of school owing to conflict with parents was higher in the smoker group than in the non-smoker group (p < .001). The percentage of OSY who used to have poor grades was higher in the smoker group than in the non-smoker group (p < .001).
Smoking behaviors based on intrapersonal, interpersonal, and community factors
Smoking behaviors according to intrapersonal, interpersonal, and community factors are shown in Table 2. The mean self-control score was higher in the non-smoker group than in the smoker group (p < .001). Concerning interpersonal factors, the parental attachment score was higher in the non-smoker group than in the smoker group (p < .001). Concerning social network, the smoking group had a higher proportion of friends (p < .001), average contact frequency (p = .002), social support frequency (p = .013), smoking support (p < .001), quality of ties (p = .002), smokers in the social network (p < .001), friends’ smoking (p < .001), and involvement of smokers in the social network (p < .001) compared with the non-smoker group. Conversely, the non-smoker group had higher general social support (p=.009) and no smoking support (p < .001). Concerning community factors, no-smoking policy exposure was higher in the smoker group than in the non-smoker group (p = .007).
Table3 and table4 are explained.
Point 7: Since OSY is influenced by many factors, it would have been great if authors included the most common factor, which is economic background. Are the youths dropping out of school due to financial constraint more prone to smoking behavior.
Response 7: When you look at table1, you can tell why the youths come to discontinue school. The 1st reason is some difficulties they have in studying, the 2nd is their lack of learning needs and the 3rd is their interaction with their out of school friends. Their lack of interest in studying is one of the major reasons students leave school while their economic status is a minor one in South Korea. Against the backdrop, it is necessary to consider creating students-friendly educational environments in which students – regardless of their aptitudes for studying – can at least complete mandatory education.
In a similar vein, we believe socioeconomic factors can have a larger impact on out-of-school youth’s smoking behavior. For more solutions, we will follow-up this issue with our future studies.

Round 2
Reviewer 2 Report
None